# NxMTransformer: Semi-Structured Sparsification for Natural Language Understanding via ADMM

**Connor Holmes**
Colorado School of Mines
Golden, CO 80401
cholmes@mines.edu

**Minjia Zhang**
Microsoft
Bellevue, WA 98004
minjiaz@microsoft.com

**Yuxiong He**
Microsoft
Bellevue WA, 98004
yuxhe@microsoft.com

**Bo Wu**
Colorado School of Mines
Golden, CO 80401
bwu@mines.edu

## Abstract

Natural Language Processing (NLP) has recently achieved great success by using huge pre-trained Transformer networks. However, these models often contain hundreds of millions or even billions of parameters, bringing challenges to online deployment due to latency constraints. Recently, hardware manufacturers have introduced dedicated hardware for NxM sparsity to provide the flexibility of unstructured pruning with the runtime efficiency of structured approaches. NxM sparsity permits arbitrarily selecting M parameters to retain from a contiguous group of N in the dense representation. However, due to the extremely high complexity of pre-trained models, the standard sparse fine-tuning techniques often fail to generalize well on downstream tasks, which have limited data resources. To address such an issue in a principled manner, we introduce a new learning framework, called NxMTransformer, to induce NxM semi-structured sparsity on pretrained language models for natural language understanding to obtain better performance. In particular, we propose to formulate the NxM sparsity as a constrained optimization problem and use Alternating Direction Method of Multipliers (ADMM) to optimize the downstream tasks while taking the underlying hardware constraints into consideration. ADMM decomposes the NxM sparsification problem into two sub-problems that can be solved sequentially, generating sparsified Transformer networks that achieve high accuracy while being able to effectively execute on newly released hardware. We apply our approach to a wide range of NLP tasks, and our proposed method is able to achieve 1.7 points higher accuracy in GLUE score than current best practices. Moreover, we perform detailed analysis on our approach and shed light on how ADMM affects fine-tuning accuracy for downstream tasks. Finally, we illustrate how NxMTransformer achieves additional performance improvement with knowledge distillation based methods.

## 1 Introduction

Large-scale Transformer networks have achieved remarkable success for a wide variety of natural language tasks, including natural language inferencing, sentiment analysis, question answering, and others. The state-of-the-art of these NLP models employs a transfer learning paradigm which contains two stages: a semi-supervised pre-training stage that trains a masked language modeling on massive web text, followed by a fine-tuning stage where the pre-trained model is adapted to specific

35th Conference on Neural Information Processing Systems (NeurIPS 2021).

downstream tasks with much smaller datasets. The size of these language models has dramatically increased in recent years; even relatively small models [6, 27] consist of hundreds of millions of parameters while larger models [23, 25, 24] stretch well into multi-billions. The large model size brings challenges for both deployment and training costs. While training a large-scale model often requires significant time even on large training clusters, the trained model also incurs significant challenges in deployment due to latency and capacity constraints.

These challenges motivate techniques to compress and accelerate these models, even in datacenter environments where hardware resource limitations are at their smallest. These techniques include but are not limited to model quantization [35, 28, 3], low rank decomposition [19], knowledge distillation [11, 27], and model sparsification [4, 9]. These compression techniques can often be combined to maximize performance gains [37, 19, 10].

Among different techniques, sparsification attempts to identify parameters that can be removed from the model without significantly compromising model accuracy. Sparsification techniques typically fall under two broad categories: unstructured and structured. Unstructured techniques will remove the individual parameters based on their importance (e.g., weight magnitude), which often yield the best accuracy but are unfriendly to modern hardware. Structured sparsification techniques remove parameters in groups (e.g., entire rows or columns), which result in models that retain their dense structure but can also add constraints that limit the expressiveness of the model.

Recently, hardware manufacturers introduced support for NxM semi-structured sparsity to provide the benefits of both structured and unstructured sparsity. In NxM semi-structured sparsity, a model may preserve M parameters from each contiguous group of N original parameters. This relatively weak constraint on sparsification allows for sparse representations similar in flexibility to those of unstructured approaches but also permits efficient hardware implementation as well. Consider the 4x2 semi-structured sparsity implementation found on NVIDIA GPUs based on the Ampere architecture [1]. The Ampere architecture introduces a small set of multiplexers that select values from the input matrix corresponding to the retained values in the weight matrix [21]. The output of this operation remains compatible with the efficient Tensor Cores for dense matrix-matrix operations. While the Ampere GPUs are the first to market with this capability, the matrix multiplication accelerators within them are similar to those used by other accelerators [15] which should enable other vendors to provide support for this type of sparsity as well.

To induce semi-structured sparsity, a small set of approaches have been offered. ASP [21] proposes training the dense network until convergence, using single-shot magnitude-based pruning to induce sparsity conformant to the NxM constraints, and repeating the original training procedure to recover accuracy. Zhou, et al. [36] uses sparse-refined straight-through estimator (SR-STE) to introduce the sparsity throughout the entire training process. Both of these techniques pose some challenges for large-scale pre-trained Transformer networks in particular. ASP can require a second costly sparse pre-train of the model and the single-shot magnitude-based pruning might hurt the knowledge transferrability to different downstream tasks. SR-STE on the other hand sparsifies from the random model initialization, which avoids the costly sparse retraining but also necessitates performing the pre-training process with only a sparse representation in mind. Since NxM sparse hardware is not yet ubiquitous, maintaining compatibility with a single dense pretrained representation is valuable so the costly pre-training process does not need to be performed for both sparse and dense networks.

In this work, we propose to effectively induce NxM semi-structured sparsity for large-scale Transformer networks to leverage newly released Sparse Tensor Core hardware by making the following contributions. (1) We introduce a principled method, NxMTransformer (See Figure 1), to obtain Transformer networks with NxM semi-structured sparsity for different downstream tasks using Alternating Direction Method of Multipliers (ADMM), a technique designed for large-scale non-convex optimization problems with constraints. Such a method allows us to alternate promoting the NxM sparsity of the network and optimizing the fine-tuning performance. (2) We conduct comprehensive experiments and demonstrate that NxMTransformer achieves 1.7 points higher accuracy than state-of-the-art techniques to introduce NxM sparsity for natural language processing tasks. (3) We perform detailed analysis on our approach and shed light on how ADMM affects fine-tuning accuracy for downstream tasks. (4) Finally, we show that NxMTransformer is complimentary to alternative model compression techniques such as knowledge distillation and can further sparsify distilled models while preserving accuracy.

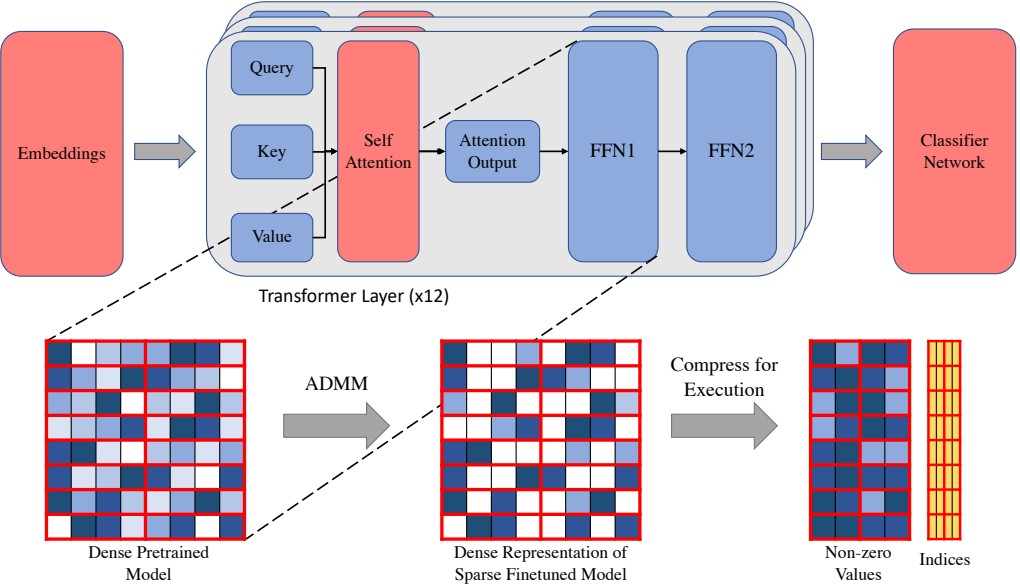

Figure 1: The layers sparsified by NxMTransformer are highlighted in blue in the block structure of a BERT model. As shown for FFN1, NxMTransformer simultaneously finetunes the pretrained representation while inducing NxM semi-structured sparsity using ADMM. This sparse model can be trivially converted to the deployment format for compatible hardware.

## 2 Background and Related Work

**Compression Techniques for NLP Models.** Due to the prominence of large-scale language models, there has been significant interest in compressing these models. Sparsification has been shown to be a promising approach to improving model inference speed and reducing memory cost earlier in computer vision [10]. Sparsifying pre-trained language models turns out to be a challenging task because the semi-supervised models tend to have less intrinsic sparsity than CNN-based models. Some prior studies try to understand and apply sparsification for large-scale language model compression. Chen, et al. [4] extend the Lottery Ticket Hypothesis [7] to pre-trained models, finding that winning tickets for the pre-training tasks do transfer universally to downstream tasks. Gordon, et al. [8] find that the level of sparsity discovered during the pretraining process represents most of the sparsity that can be discovered in the model. Guo, et al. [9] show that a proximal pruning strategy achieves higher accuracy than competing lasso regularization methods and iterative magnitude pruning. Most of these studies still focus on unstructured sparsity, which encounters difficulty to obtain large speedups on modern hardware. As a structured technique, Michel, et al. [20] observe that many transformer heads themselves are redundant and can be pruned with minimal accuracy loss. While a structured technique, this technique provides limited acceleration benefits since much of the compute time for short-to-medium sequence lengths is spent in the intermediate layers rather than the attention heads themselves. In addition to parameter sparsification, BERT models can also realize compression gains with model quantization [35, 28] and knowledge distillation [27, 29, 30, 14]. These techniques are complimentary to our proposed method and can be combined to achieve more effective overall compression, as in [10].

**Semi-structured Sparsity.** There have been few studies to induce NxM sparsity for DNN models. Among them, SR-STE [36] is the most comparable work to this paper. SR-STE is single stage compression technique that uses an extended Straight-through Estimator and a sparse-refined term to improve the induction of NxM sparsity in the network. SR-STE is demonstrated across image and neural machine translation tasks and shows improved accuracy over NVIDIA's ASP [21]. Unlike this work, SR-STE is designed for training from random initialization, not a pretrained model.

From a performance perspective, cuSPARSElt [21] (the NVIDIA linear algebra library for NxM sparsity) demonstrated 1.3X to 1.6X performance uplift for the matrix multiplications prevalent in BERT models using sparse NxM Tensor cores on Nvidia A100 GPUs. Higher speedups may be

achieved by increasing batch size, sequence length, or model dimensionality with a maximum of 1.8X improvement for fp16 and 1.9X for int8. Realized speedups are smaller than the theoretical 2X improvement of 50% sparsity primarily due to memory bandwidth constraints. The performance provided by cuSPARSElt is a reasonable expectation for what can be achieved with NxMTransformer.

Models including NxM sparsity may be deployed for serving using optimized frameworks like TensorRT [2], which include optimized layer operators that take advantage of the hardware's features. Moving forward, inference-driven machine learning compilers like TVM [5] can also introduce support for the semi-structured sparse Tensor operators using the same mechanisms as were used to provide support for dense Tensor cores, which use a similar API.

**ADMM for Neural Networks.** Alternating Direction Method of Multipliers (ADMM) has been used in previous works primarily for compressing convolutional neural networks. Ye, et al. [34] explore an iterative approach to ADMM that shortcuts full ADMM convergence with a masked retraining operation. Ren, et al. [26] use ADMM for both unstructured sparsification and quantization of the model while combining the technique with a heuristic for determining whether speedup will be achieved given the achieved compression of a layer. Ma, et al. [18] use domain knowledge of CNN filter structure to perform a structured pruning operation for mobile-hardware efficient compression. While the above techniques all use ADMM as the sparsifying mechanism, none examine in-depth applicability to pre-trained language models for NxM sparsity.

## 3 Methodology

In this section, we formulate ADMM for NxM sparsity, describe the specific aspects of pre-trained models that define the optimization pipeline, and describe the high-level optimization schedule.

### 3.1 Problem Definition.

Consider adapting a pre-trained large-scale language model $\Phi$ with $L$ Transformer layers (e.g., BERT) to natural language understanding tasks, such as sentiment analysis, entailment, question and answering, etc, where the training instances are inputs (often text phrases) and target pairs: $\{x_i, y_i\}_{i=1}^N$. Assume the collection of pre-trained model weights is $W^0 = \{\mathbf{W^0}_i\}_{i=1}^L$[1], the goal of NxMTransformer is to load $W^0$ and fine-tune it to $W'$ such that $\{W'_i\}_{i=1}^L$ satisfies the constraints of at most $M$ weight parameters having non-zero values out of $N$ consecutive weights, while achieving similar performance in comparison to fine-tuning the task-specific objective function $f(\{\mathbf{W}_i\}_{i=1}^L)$ (e.g., cross-entropy for classification) using $W_0$ but without constraints.

### 3.2 NxMTransformer

Different from conventional DNN training objectives, the above problem is non-convex with combinatorial constraints. Therefore, it cannot be directly solved by gradient-based methods such as stochastic gradient descent. To address this issue, we adopt the alternating direction method of multipliers (ADMM), which is a reliable method for large-scale constrained optimization (e.g., with combinatorial constraints). In particular, we modify the objective function of the NxM sparsity problem as

$$\min_{\{\mathbf{W}_i\}} f(\{\mathbf{W}_i\}_{i=1}^L) + \sum_{i=1}^L g_i(\mathbf{Z}_i) \text{ subject to } \mathbf{W}_i = \mathbf{Z}_i, i = 1, \ldots, L \tag{1}$$

where $f(\cdot)$ is the fine-tuning objective function, and $g_i(\cdot)$ is an added penalty function and $Z_i$ are auxiliary variables. To apply ADMM, we define the penalty function as

$$g_i(\mathbf{W}_i) = \begin{cases} 0 & \text{if } \mathbf{W}_i \in \mathbf{S}_i \\ \infty & \text{otherwise} \end{cases} \tag{2}$$

---

[1] For convenience's sake, we will omit the notation of the bias since it is not relevant to the task of sparsification.

where $\mathbf{S}_i$ represents the constraint set $S_i = \{W_i$ that have at most M weight parameters having non-zero values out of N consecutive weights$\}$.

**Choice of $S_i$.** Not all weights in pre-trained Transformer models need to satisfy this NxM sparsity constraint. BERT and similar pretrained models typically consist of three major components: embeddings, Transformer blocks, and classifiers (See Figure 1). For NxM semi-structured sparsity, we solely consider weights in Transformer blocks. Take BERT as an example, each Transformer block consists of 6 fully connected sub-layers: the query $Q$, key $K$, value $V$ layers, the attention output layer $Attn.$, and two feed-forward network $FFN1$ and $FFN2$. Each of the fully connected layers can take advantage of NxM semi-structured sparsity; furthermore, these layers constitute the vast majority of inference wall-time. Of the six fully connected layers, $FFN1$ and $FFN2$ are particularly important to sparsify, alone requiring more than half of the inference wall-time for a Transformer block. Note that attention head pruning techniques [20] are unable to sparsify $FFN1$ and $FFN2$. The self-attention mechanism itself does not include any trained parameters and is unable to be sparsified using this technique. We exclude the embedding layer since the lookup operation associated with the embedding layers is incompatible for acceleration with Tensor Cores. The classifier is composed of fully connected layers. For a given task, the classifier weights are randomly initialized at the beginning of the fine-tuning process. We find that sparsifying these matrices using ADMM will unnecessarily harm accuracy. Since the execution of these layers is typically under 2% of the inference wall-time, the runtime cost is minimal for doing so.

**Decomposing the minimization problem into sub-problems.** Once we define $S_i$, we apply the augmented Lagrangian, which decomposes equation 1 into two sub-problems on $W$ and $Z$:

$$\text{Sub-problem 1 (\textbf{performance-promoting}):} \min_{\{\mathbf{W}_i\}} f(\{\mathbf{W}_i\}_{i=1}^L) + \sum_{i=1}^L \frac{\rho}{2}\|\mathbf{W}_i - \mathbf{Z}_i^k + \mathbf{U}_i^k\|_F^2 \quad (3)$$

$$\text{Sub-problem 2 (\textbf{NxM sparsity-promoting}):} \min_{\{\mathbf{Z}_i\}} \sum_{i=1}^L g_i(\mathbf{Z}_i) + \sum_{i=1}^L \frac{\rho}{2}\|\mathbf{W}_i^{k+1} - \mathbf{Z}_i + \mathbf{U}_i^k\|_F^2 \quad (4)$$

The first sub-problem solves the *performance promoting* problem, which consists of two terms. The first term is the standard objective function for fine-tuning the task, and the second term is a $L_2$ regularization term. The regularization target $Z_i^k - U_i^k$ is dynamically updated, based on $U_i^k$, which is the dual variable (i.e., the Lagrange multiplier). Since the $L_2$ term is convex, the complexity of solving sub-problem 1 (e.g., via ADAM [16]) is the same as minimizing $f(\cdot)$. The second sub-problem solves the *sparsity promoting* problem. Since it optimizes the sparse constraints separately, it can be solved analytically as the solution of the Euclidean projection of $\mathbf{W}_i^{k+1} + \mathbf{U}_k$ onto our constraint. For the case of NxM semi-structured sparsity, this is accomplished by retaining the M largest values of the contiguous group of N values (See Figure 1), which can be solved in linear time. Finally, we need to update the dual variable $U$ as $\mathbf{U}_i^k := \mathbf{U}_i^{k-1} + \mathbf{W}_i^k - \mathbf{Z}_i^k$ to guarantee that the dual feasibility condition is satisfied in each ADMM iteration.

**Sparsity-inducing based fine-tuning.** The typical ADMM pipeline fully trains a model to convergence before introducing ADMM [26]. This two-step process is necessary since the primary objective of ADMM is to optimize the existing network to conform to the constraints; ADMM will only introduce small changes to parameters it retains in the model. However, for pre-trained language models, the primary purpose of the fine-tune is to adapt the classifiers for the specific downstream with minimal disturbance to the parameters of the pre-trained representation. As a result, we apply the three aforementioned steps (i.e., two sub-problems and the update of the dual variable) while fine-tuning the model. In particular, we perform the three steps in an alternating manner, i.e., performing some number of fine-tuning steps with Adam to solve the first sub-problem, solving the Euclidean projection for each weight matrix for the second sub-problem, and finally updating the auxiliary variable. This sequence will be referred to as one ADMM iteration. The optimization proceeds until the $\mathbf{W}$ and $\mathbf{Z}$ variables have converged, at which point we have a sparsified network compliant with our NxM constraint.

# 4 Evaluation

In this section, we evaluate NxMTransformer and show its effectiveness in compressing Transformer networks over a wide range of NLP tasks.

**Implementation.**  NxMTransformer is implemented as a PyTorch [22] compatible library for sparsifying models with NxM semi-structured sparsities. Furthermore, a HuggingFace Transformers [33] compatible Trainer is implemented to enable easy integration with their model collection and training scripts. Our approach supports different NxM sparse patterns (e.g., 4:1, 8:4) so long as the weight's input dimension is a multiple of N. For evaluation, we focus on evaluating 4:2 sparsity since it is supported in commodity hardware. We use pretrained model checkpoints for both BERT[2] and DistilBERT[3], provided by the HuggingFace model repository. All models were fine-tuned on an Intel Xeon 2630 v4 server with 2x NVIDIA Titan V running Ubuntu 18.04. PyTorch version 1.7.1 was used alongside Transformers 4.3.2. Finetuning these models required between 5 minutes (RTE) and 5 hours (MNLI) depending on task size. For the set of training hyperparameters used for training NxMTransformer, see Table 3.

**Dataset.**  We evaluate NxMTransformer and our baselines using the the General Language Understanding Evaluation (GLUE) benchmark [31], a collection of NLP tasks varying in data availability and complexity. We report the Spearman correlation for STS-B, the F1 score for MRPC, Matthews correlation for CoLA, and accuracy for all remaining tasks. The reported average is the geometric mean of reported scores.

## 4.1 Main Results

- **BERT** [6]: This is the $BERT_{base}$ model from publicly available checkpoint.
- **ASP**: Inline with ASP practices[21], we perform one-shot magnitude-based masked pruning on the pretrained model. This baseline is considered best practices for a large pretrained language representation for semi-structured sparsity.
- **ADMM_Unstructured**: To measure the accuracy cost of semi-structured accuracy specifically, we create another baseline that uses ADMM but induces unstructured sparsity at 50% per-layer (rather than global) sparsity.

**Hyperparameters.**  In [6], the authors only report the development results on a few tasks. Therefore, we produce the BERT baseline results. We fine-tune BERT for 5 epochs on each downstream task. We perform a grid search of batch sizes 16 and 32, and learning rates 1e-5, 3e-5, and 5e-5 for SST-2, QNLI, and MNLI, due to their high training cost. Learning rates of 7e-5 and 9e-5 are additionally used for the remaining tasks. For masked fine-tune, the model was fine-tuned with learning rates 1e-5, 3e-5, 5e-5, 7e-5, and 9e-5 across batch sizes 16 and 32. ADMM_Unstructured is trained using the same hyperparameters sweeps as NxMTransformer. For all configurations, we set the fine-tune to have 5 epochs, and the best observed result on the validation set is reported.

We report the evaluation results for BERT in Table 1 and make the following key observations.

First, the pruning based method sparsifies weights of Transformer blocks but cannot explicit satisfy the underlying hardware constraints, e.g., the 4:2 sparsity. Although preserving the highest accuracy on downstream tasks (81.3 vs. 81.8 on average), the obtained sparse weights have a random structure of non-zero weights, which is inefficient to execute in modern hardware systems. As a result, the performance benefit with these unstructured sparsity based approaches is negligible, even when the pruning rate is high (e.g., 95%) [32].

Second, when it comes to NxM sparsity, NxMTransformer achieves an average score of 80.4, outperforming ASP by 1.6 points. In particular, we observe that for large tasks (MNLI, QNLI, SST-2), NxMTransformer performs comparably to ASP. However, NxMTransformer dramatically outperforms ASP for the small tasks (CoLA, STS-B, MRPC, RTE), increasing accuracy by 2.9 points on average. Since the smaller tasks can be more sensitive to random seed performance, we also performed a random seed sweep across 7 random seeds. NxMTransformer still outperformed

---

[2] https://huggingface.co/bert-base-uncased, Apache 2.0 License
[3] https://huggingface.co/distilbert-base-uncased, Apache 2.0 License

Table 1: The dev set results on the GLUE benchmark. The results show that NxMTransformer is able to achieve higher accuracy than ASP for NxM sparsity, especially when the downstream tasks have low data resources.

| Model | Task | | | | | | | Average |
|---|---|---|---|---|---|---|---|---|
| | MNLI (m/mm) | SST-2 | QNLI | CoLA | STS-B | MRPC | RTE | |
| Samples | 392k | 67k | 108k | 8.5k | 5.7k | 3.5k | 2.5k | |
| Baseline (BERT$_{base}$) | 84.5/84.8 | 92.5 | 91.6 | 56.7 | 89.6 | 91.7 | 70.7 | 81.8 |
| ADMM$_{Unstructured}$ | 84.0/84.7 | 92.5 | 91.0 | 57.5 | 89.6 | 90.5 | 68.2 | 81.3 |
| ASP | **83.3**/83.4 | 91.9 | **90.6** | 51.7 | 88.7 | 88.1 | 63.9 | 78.8 |
| NxMTransformer | 82.3/**83.4** | **92.3** | 90.4 | **55.3** | **89.3** | **90.8** | **68.6** | **80.5** |

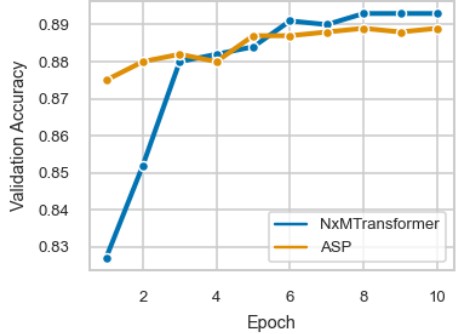

(a) Validation accuracy of NxMTransformer and ASP networks on best configuration STS-B.

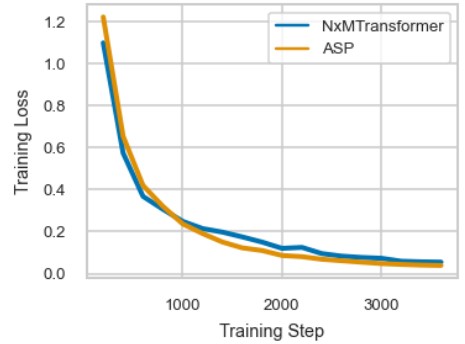

(b) Training loss of the NxMTransformer and ASP networks on best configurations of STS-B.

Figure 2: ASP and NxMTransformer on STS-B

ASP by 1.9 points across the tasks with fewer than 10,000 training examples when median was our metric of accuracy rather than maximum. This pattern suggests that while more data allows a less principled mechanism to recover accuracy, an explicit optimization approach that take the sparsity constraints into account would yield much better accuracy results when the downstream tasks have low data resources. As a result, NxMTransformer retains 99% of the accuracy of the unstructured sparsity (ADMM$_{Unstructured}$) and 98.4% of the uncompressed model (BERT$_{base}$). Different from ADMM$_{Unstructured}$, which suffers from expensive irregular memory accesses, our NxMTransformer method can effectively leverage the underlying Sparse Tensor Core and achieves inference speedups even with 50% overall sparsity.

## 4.2 Analysis Results

In this section, we further investigate the performance gain of NxMTransformer and its impact to the downstream task accuracy with NxM sparsity.

**Validation accuracy improvement.** We first compare the model trained with ASP and NxMTransformer. To evaluate NxMTransformer, we perform a hard prune of small weights at the end of every epoch, so we evaluate the model as if it has already sparsified. As we can see in Figure 2, NxMTransformer converges slower than ASP in the beginning of the fine-tuning. This is presumably because the initial model weights are heavily violating the hardware constraints, causing significant degradation when performing the pruning action. As the training moves forward, NxMTransformer is able to catch up and outperform ASP at around epoch 6. This is because by using ADMM, NxMTransformer trains the dense model to gradually converge to a sparse model that satisfies the provided constraints, so pruning weights would gradually have a smaller impact to model accuracy. Since then, the validation accuracy of both ASP and NxMTransformer are still increasing, but ASP tends to plateau after 8 epochs, whereas NxMTransformer continues to increase, outperforming ASP by 0.5 point in the end. On the other hand, we also observe that ASP has slightly lower training loss

towards the end (as shown in Figure 2b), indicating that ASP might be overfitting on the dev set (potentially due to the small amount of data).

**Dynamism of sparse subnetwork masks.** The penalty parameter $\rho$ controls the balance between maximizing the accuracy of the model and reaching a sparse model. The larger that $\rho$ is, the greater the influence of the sparsity-inducing regularizer and the more quickly the model converges to a sparse solution. The trade-off represented by tuning $\rho$ manifests itself by values moving into and out of the sparse subnetwork mask between ADMM iterations. Since the sparsity is induced more slowly by a small $\rho$, a parameter is more likely to be included when the second sub-problem is solved. Empirically, we find that frequently moving values into and out of the sparse subnetwork mask results degrades the ultimate sparse networks accuracy. In Figure 3, $\rho$ is tuned to achieve different values of similarity, which is calculated as the fraction of values that remain in the sparse subnetwork mask from one ADMM iteration to the next. For both CoLA and QNLI a clear correlation between average similarity and accuracy exists until approximately 99% similarity, where the strength of the regularizer is over-weighted against the training loss and accuracy begins to degrade.

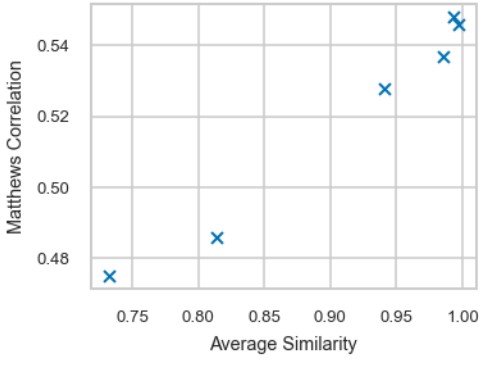
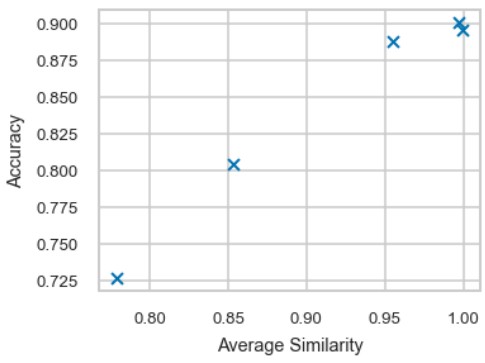

(a) Similarity vs Matthews Correlation on CoLA (8.5k). Learning rate: 9e-5, Batch size: 32

(b) Similarity vs Accuracy on QNLI (108k samples). Learning: 3e-5, Batch size: 32

Figure 3: Analysis of mask similarity and accuracy metrics for a subset of GLUE tasks. Different similarities are achieved through $\rho$ tuning.

Inspecting the values of weights that undergo swapping illustrates why higher dynamism in the sparse subnetwork mask incurs an accuracy penalty. Figure 4 shows a parameter outside of the sparse subnetwork mask for just a single iteration will decrease in magnitude by approximately 15% from its initial magnitude. A second iteration further increases this penalty to 25%. This is in contrast to parameters that remain in the sparse mask for the entirety of the optimization process and retain all of their magnitude. The philosophy behind training from a pretrained model is to retain the information of that process. Large changes in parameter magnitude are destructive to that pretrained information because the parameter only partially reflects that learned information.

### 4.3 When NxM Sparsity Meets Knowledge Distillation

Knowledge distillation has been proven to be another promising technique to compress a large model and also yield models with regular and dense structures. However, there have been few studies on the sparsity of knowledge distilled models, especially in the context of transfer learning and pre-trained language models. On first sight, it may seem that once distilling a large model into a smaller model, there will be less redundancy in the model, where sparsification might hurt model accuracy significantly. In this section, we investigate how NxMTransformer affects KD compressed models. We apply NxMTransformer to a student model obtained through DistilBERT [27], which is a 6-layer BERT model with 768 hidden dimension size. The results are shown in Table 2, and we make the following observations.

First, despite DistilBERT and NxMTransformer have the same number of parameters, NxMTransformer achieves 2 points higher accuracy on average than DistilBERT, which indicates that removing

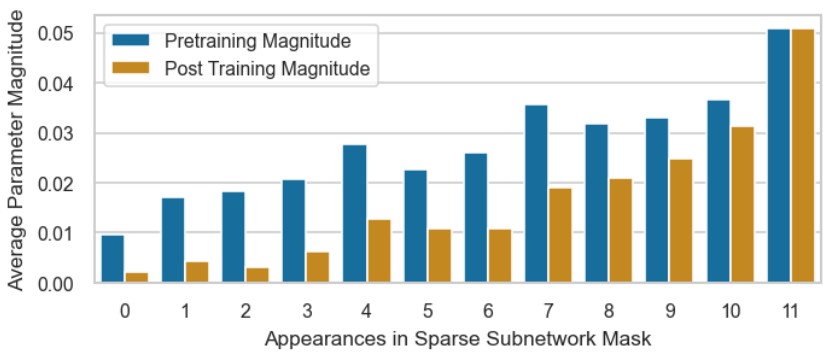

Figure 4: Comparison of average parameter value before and after fine-tuning on a 10-epoch STS-B experiment (learning rate: 5e-5, batch size: 16, $\rho$: 1e-3) based on the number of times it was present in the sparse subnetwork mask.

Table 2: The dev set results on the GLUE benchmark with knowledge distillation. The results show NxMTransformer retains 97.6% of the DistilBERT model.

| Model | # Params | Task | | | | | | | Average |
|---|---|---|---|---|---|---|---|---|---|
| | | MNLI (m/mm) | SST-2 | QNLI | CoLA | STS-B | MRPC | RTE | |
| NxMTransformer (4:2 BERT$_{12}$) | 66.6M | 82.3/83.4 | **92.3** | **90.4** | **55.3** | **89.3** | **90.8** | **68.6** | **80.5** |
| DistilBERT (BERT$_6$) | 66.6M | **82.4**/82.5 | 90.9 | 89.1 | 53.4 | 86.6 | 89.6 | 63.5 | 78.5 |
| DistilBERT-NxMTransformer | 45.3M | **80.7**/81.2 | 90.5 | 87.5 | 50.1 | 87.1 | 88.7 | 59.2 | 76.6 |

Transformer layers from the BERT model is more detrimental to the model accuracy and NxMTransformer's semi-structured approach captures redundancy (intra-layer) much more efficiently.

Second, NxMTransformer retains 97.6% of the accuracy of the dense DistilBERT model. While slightly worse than the retained accuracy ratio for BERT$_{base}$ (98.4%) itself, this indicates that while the depth of dimensionality of the model may be reduced, the relatively low amounts of sparsity exploited by semi-structured sparsity are still prevalent in fully connected layers. The result also seems to suggest the potential existence of a winning ticket even in highly compressed BERT model [4].

More recently, knowledge distillation techniques such as TinyBert [14] and MiniLM [32] leverage fine-grained knowledge transfer to help student better mimic teacher's behavior. As our method is largely orthogonal to how knowledge gets transferred between teacher and student, we expect the effectiveness on NxMTransformer as witnessed on DistilBERT should apply to models distilled through these more advanced techniques as well and will leave more extensive studies as future work.

## 5   Conclusion

Semi-structured sparsity can improve runtime resource efficiency without large penalties in model performance. This work demonstrates the effectiveness of a low-overhead ADMM approach to introduce NxM semi-structured sparsity for large pretrained natural language models. Furthermore, NxMTransformer is an orthogonal optimization to existing compression techniques, such as knowledge distillation and reduced precision inference. However, NxMTransformer is limited in that it is not a lossless compression technique and does introduce an accuracy gap. Furthermore, it is untested on emerging pretrained Transformer representations for vision tasks and it is unclear how it would transfer to this emerging domain.

## 6   Negative Societal Effects

NxMTransformer exposes two key avenues for negative societal impacts. First, since NxMTransformer is designed to inherit from a pretrained model representation, it inherits any societal-level biases that may be embedded in the parent model. Previous work has identified that BERT models do

encode both gender bias [17] and bias against people with disabilities [13]. NxMTransformer does not specifically attempt to combat these biases and downstream tasks fine-tuned with NxMTransformer will inherit them as well. The second potential source of negative societal impacts is due to the act of pruning itself. Hooker, et al. [12] identify that for convolutional neural networks, pruning can disproportionately reduce accuracy of lower frequency output examples. Although the model design for CNNs is different from that of Transformers, it is unlikely this alone would mitigate this source of network bias. These sources of bias can introduce real-world harms as fine-tuned natural language models are increasingly used for online content moderating, brand sentiment, career matching, and other human-facing algorithms that can affect livelihoods.

## Acknowledgments and Disclosure of Funding

We thank the anonymous NeurIPS reviewers for their constructive comments. This work was supported by the National Science Foundation under grants CCF-1823005 and an NSF CAREER Award (CNS-1750760).

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
