# A  Appendix

## A.1  Scheduling ADMM Iterations

ADMM can converge effectively with as few as 80 training steps in each ADMM iteration. For example, RTE (2.5k training samples), ADMM successfully converges with a minibatch of 32 and one ADMM iteration per epoch. However, increasing the number of training steps between iterations can reduce reliance on a high learning rate. Note that a high learning rate is necessary to allow the optimizer to relatively quickly push larger parameters towards 0 in a reasonable number of training steps, since the practical parameter delta in a single training step is proportional to the product of the learning rate and $\rho$. Furthermore, a small learning rate reduces the effectiveness of the regularizer and decreases model similarity.

Experimentally, ADMM will achieve its maximum accuracy once 10 ADMM iterations have occurred. However, further optimizing, does not appear to harm model accuracy. While further training is typically not desirable for small tasks — training is frequently extended for these tasks to have a sufficiently large training period each ADMM iteration — for large tasks tens of ADMM iterations may be performed such that the fine-tune can continue for sufficient time. For example, a fine-tune on QNLI for just 3 epochs may perform nearly 50 ADMM iterations.

Table 3: NxMTransformer Training Hyperparameters. Smaller tasks utilize larger learning rates and penalty parameters ($\rho$) since ADMM iterations for these tasks are much shorter (See Appendix A.1).

| Tasks | Learning Rates | $\rho$ | Batch Size | Epochs |
|---|---|---|---|---|
| MNLI, QNLI, SST-2 | 1e-5, 3e-5, 5e-5 | 4e-4, 1e-3, 3e-3 | 16, 32 | 5 |
| CoLA, STS-B, MRPC, RTE | 5e-5, 7e-5, 9e-5, 1e-4 | 3e-3, 6e-3, 1e-2 | 16, 32 | 10 |