# OpenReview forum: "NxMTransformer: Semi-Structured Sparsification for Natural Language Understanding via ADMM"
_NeurIPS.cc/2021/Conference — NeurIPS 2021 Poster_

### Official Review · Reviewer_NSKn · 2021-07-16

**Rating:** 6
**Confidence:** 4

**Summary:**

The paper proposed to apply the recently developed NxM semi-structured sparsity by hardware manufacturers to compress BERT, with an ADMM-based sparsifying mechanism. Specifically, the authors used 4x2 semi-structured sparsity pattern since it's supported by NVIDIA GPUs, which halves the number of parameters. In the experiments, the paper showed that the ADMM-based approach outperforms the previous ASP method, which requires a second costly sparse pre-train of the model. The proposed model also demonstrates better performance compared to DistilBERT under the same number of parameters setting.


**Limitations And Societal Impact:**

Yes

**Main Review:**

Advantages:
1. The proposed method show good empirical performance when compared to other baseline models such as ASP or DistilBERT.
2. The paper is the first to apply ADMM-based sparsifying mechanism to pre-trained language models.

Weakness:
1. Given that both NxM semi-structured sparsity and ADMM-based sparsifying mechanism are already well-developed, this paper only combines these two techniques and thus lack novelty.
2. One of the advantages of NxM semi-structured sparsity compared to unstructured sparsity is its superior inference speed. However, the paper didn't present any experimental results w.r.t. inference speed or latency.
3. In "When NxM Sparsity Meets Knowledge Distillation", the paper might also need to present a DistilBERT model with 45.3M parameters to show the performance gain of DistilBERT-NxMTransformer. Otherwise, it's unclear how DistilBERT-NxMTransformer is better than other baselines under the same number of parameters setting.


**Time Spent Reviewing:**

2

---

> ### Author Response · Authors · 2021-08-10
> **Addressing Novelty, Inference Benefit, and Distillation Concerns**
>
> Thank you for taking the time to review our work.
>
> We believe the novelty in the work derives from how our ADMM approach induces NxM sparsity to these models much more cheaply than existing techniques. The cited ASP work provides lower accuracy when applied directly to the fine-tuning process and requires a full repetition of the pre-training phase when applied more aggressively. In contrast, SR-STE requires random initialization and the full pre-training phase. A second novel contribution is our investigation into the mechanics of how ADMM induces sparsity. No previous works have attempted to identify the effects of the ADMM hyperparameters on mask dynamism or parameter value. In fact, some previous works suggest that ADMM’s hyperparameters are rather insensitive and don’t require tuning, which our work refutes.
>
> For the potential for acceleration, we cite cuSPARSElt’s (Section 2, Semi-structured Sparsity) improvement against dense matrix multiplications as reasonable expectations for inference acceleration. Since NxMTransformer is only changing how the sparse model is achieved, its performance will be identical to cuSPARSElt. TensorRT 8, which was recently released, demonstrates performance gains consistent with those demonstrated by cuSPARSElt as well.
>
> We don’t include a 45.3M parameter DistilBERT comparison because DistilBERT checkpoints require pre-training distillation and there is no publicly available checkpoint for this dimensionality. We do believe the results as presented do demonstrate that there is no significant interference between the two techniques and that NxM sparsity and distillation can be cooperatively combined for additive compression.

---

> > ### Comment · Reviewer_NSKn · 2021-08-19
> > **Official Comment**
> >
> > I have read other reviews and the authors' rebuttal. I appreciate the authors' detailed explanations on paper novelty and speed performance, which addressed my concerns. As for DistillBERT-NxMTransformer, I still believe that a baseline model with 45.3M parameters is necessary. Otherwise, it is unclear to me that what conclusions can be drawn from Section 4.3 and I would suggest the authors to remove this subsection.
> >
> > Overall, I will increase my score from 4 to 6.

---

### Official Review · Reviewer_k8jW · 2021-07-16

**Rating:** 6
**Confidence:** 4

**Summary:**

This paper introduces a new method called NxMTransformer to induce NxM semi-structured sparsity on pertained language models, for recent hardware dedicated for efficient NxM sparsity (e.g. newly released Sparse Tensor Core). NxMTransformer fromulates the NxM sparsity as a constrained optimization problem and use ADMM to alternately promote the NxM sparsity constraint and finetuning performance. Experiments on GLUE datasets show that the proposed approach can beat the best NxM sparsity method by 1.7 points.


**Limitations And Societal Impact:**

Yes.

**Main Review:**

Pros:
* Developing parameter-efficient NLP models is an important research direction. It is especially cool that this paper considers efficient NLP methods optimized for specific hardwares.
* This paper introduces a novel method to induce NxM sparsity for recently released hardware and beats the state-of-the-art NxM sparsity method on GLUE datasets.

Cons:
* The experimental results can be stronger
    - Currently, it only evaluates on GLUE validation dataset with BERT-base models, and it only considers one parameter-efficient baseline ASP. It can potentially evaluate its method on GLUE test sets with BERT-large models, and consider more parameter-efficient methods as baselines (even if they are not all NxM sparsity).
     -  Many parameter-efficient NLP methods (e.g. see missing references) can achieve <1% sparsity (compared to 50% sparsity in this paper) on GLUE datasets without losing much accuracy -- even though those methods are not NxM, the paper will benefit if authors can compare and justify its method's advantages over those methods.
    -  This paper currently beats ASP mostly on the small datasets. Those small datasets are very small and can have high variance even with hyper-parameters tuning. It will be good to justify its advantages on more datasets.
* Missing background and importance of NxM structured sparsity
   - What is NxM sparsity? How does NxM sparsity hardware work?
   -  What hardware will it exactly be useful for (newly released Tensor Cores?), and why are that hardware important?
   - What are the practical benefits to satisfy NxM sparsity? Does it improve inference speed, by how much?

Question
        - Can a similar approach (ADMM pipeline) be used to optimize for another type of sparsity / hardware?

Missing References
  * Mahabadi et al, Parameter-efficient Multi-task Fine—tuning for Transformers via Shared Hypernetworks
  * Guo et al, Parameter-efficient transfer learning with Diff Pruning
  * Mahabi et al, Compactor: Efficient Low-Rank Hypercomplex Adapter Layers
  * Zaken et al, BitFit: Simple Parameter-efficient Fine-tuning for Transformer-based Masked Language-models

Update: I've decided to keep the scores after reading the latest responses.



**Time Spent Reviewing:**

1

---

> ### Author Response · Authors · 2021-08-10
> **Addressing Related Literature, Small Task Variability, and Technique Flexibility**
>
> Thank you for taking the time to review our work.
>
> In your review, you’ve identified a series of research works for parameter-efficient training for transfer learning. We believe this work is separate from the presented work on NxM sparsity because parameter-efficient training techniques aim to reduce the training cost of transfer learning for a dense model whereas NxM sparsity (and semi-structured sparsity in general) are geared towards improving inference latency. Since these goals are orthogonal from each other, we determined it was unnecessary to include them as a separate baseline.
>
> As you note, NxMTransformer provides greater accuracy gains across the smaller tasks, which can show higher sensitivity to hyperparameters and random seed. We have performed additional trials across the small configurations and have measured similar sensitivity to random seed for both ASP and NxMTransformer. Across the tasks with fewer than 10,000 training examples, NxM Transformer still provides 1.9 points of accuracy increase on average (1.9% on CoLA, 2.2% MRPC, 3.3% RTE, .3% STS-B) when comparing the respective medians of the random seed sweep. We believe the consistent accuracy gain across this random sweep demonstrates NxMTransformer’s superior accuracy is an artifact of the technique rather than random noise.
>
> The NxMTransformer ADMM pipeline can be used to flexibly target different types of hardware. We present 4:2 semi-structured sparsity since it is currently the only hardware in production using this sparsity at this time but adapting to different types of sparsity would only require changing the constraints for the projection in the second sub-problem of ADMM. For example, adapting to 16:4 semi-structured sparsity would change the projection to retain the 4 largest values of each 16 in a vector rather than the 4 and 2 presented. Non-vector sparsity can be supported in a similar manner. This is possible since ADMM itself is a general optimization technique; if the sparsification constraints can be represented in the form of the indicator function, then it should be feasible to integrate into the pipeline.
>
> NxM sparsity is the retention of M values from N consecutive values in the dense model to produce a sparse model. So for 4:2 NxM sparsity, we retain 2 out of every group of 4 values. When we compress the model, we can then only store the two retained parameters, as well as their indices within each group of 4 (so 00, 11 would encode keeping the first and last values in a group). When executed on hardware, the hardware will load an input tile of the input matrix into local memory and then use the indices to select only the columns that are necessary for each part of the dot product. This style of compression reduces both memory bandwidth required and the amount of operations the ALUs need to perform. For further information, the NVIDIA A100 Whitepaper (https://images.nvidia.com/aem-dam/en-zz/Solutions/data-center/nvidia-ampere-architecture-whitepaper.pdf) has documentation of how Sparse Tensor Cores function on Page 31.

---

### Official Review · Reviewer_KcKK · 2021-07-18

**Rating:** 7
**Confidence:** 4

**Summary:**

This paper describes a ADMM-based approach to NxM sparsification in large-scale language models (e.g. BERT). Semi-structured sparsification, such as the 4:2 sparsity implementation of Ampere GPUs, allows for a best of both worlds: the weak constraint (picking M weights out of a contiguous N) enables sufficient expressivity to not overly degrade accuracy, while still allowing for an efficient hardware implementation. The authors compare against a prior approach (ASP) on a range of GLUE benchmarks (MNLI, SST-2, QNLI, CoLA, STS-B, MRPC, and RTE), outperforming ASP by 2.9 points on average for tasks with small datasets (and comparably on those with larger datasets). The authors also provide insight into the fine-tuning accuracy, and show that the approach is complementary to a (recent but not quite state-of-the-art) knowledge distillation method.

**Limitations And Societal Impact:**

I was satisfied with the authors' statement on limitations and societal impact.

**Main Review:**

The target problem is interesting with high potential impact, especially given the rising prevalence large language models.

The novelty is not particularly high, given that ADMM has already been applied to the problem of compressing neural networks, and similar previous approaches have been proposed for NxM sparsification, albeit for randomly initialized models (rather than pretrained ones).

The authors performed quite extensive evaluation on GLUE benchmarks, but the reported improvements were generally quite modest. It was interesting to see empirically that this approach is in fact orthogonal to a form of knowledge distillation.

The quality of writing was not overly high, making it difficult to understand the paper in places. In addition to clarifying explanations, the authors should clean up various editing issues:
(1) Remove "(needs CCITE)" from the first sentence of the introduction.
(2) "a masked language modeling" -> "model"
(3) "NxMTransformer(See Figure 1") -> Add space
(4) "Finally, we show that NxMTransformer is complimentary to" -> "complementary"
(5) Many strange citations of the form "[LASTNAME], et al." -> remove the comma here
(6) "for the specific downstream with minimal disturbance" -> "downstream task"
(7) "from publicly available checkpoint" -> "from a publicly available checkpoint"
(8) "Inline with" -> "In line with"
(9) "The baseline is considered best practices" -> "best practice"
(10) "using the same hyperparameters sweeps" -> "hyperparameter"
(11) "Different from" -> "Unlike"
(12) "moving values into and out of the sparse subnetwork mask results degrades the ultimate sparse networks accuracy" -> remove "results" and "networks" -> "network's"
(13) "despite DistilBERT and NxMTransformer have the same number of parameters" -> "having"

**Time Spent Reviewing:**

4

---

> ### Author Response · Authors · 2021-08-10
> **Addressing Novelty Concerns**
>
> Thank you for taking the time to review our submission.
>
> We believe the novelty in the work derives from how our ADMM approach induces NxM sparsity to these models much more cheaply than existing techniques. The cited ASP[1] work provides lower accuracy when applied directly to the fine-tuning process and requires a full repetition of the pre-training phase when applied more aggressively. In contrast, SR-STE[2] requires random initialization and the full pre-training phase. A second novel contribution is our investigation into the mechanics of how ADMM induces sparsity. No previous works have attempted to identify the effects of the ADMM hyperparameters on mask dynamism or parameter value. In fact, some previous works suggest that ADMM’s hyperparameters are rather insensitive and don’t require tuning, which our work refutes.
>
> Thank you for identifying those styling errors and we will work to correct them as well.
>
> [1] Asit Mishra, Jorge Albericio Latorre, Jeff Pool, Darko Stosic, Dusan Stosic, Ganesh Venkatesh, Chong Yu, and Paulius Micikevicius. Accelerating sparse deep neural networks, https://arxiv.org/abs/2104.08378, 2021.
>
> [2] Aojun Zhou, Yukun Ma, Junnan Zhu, Jianbo Liu, Zhijie Zhang, Kun Yuan, Wenxiu Sun, and Hongsheng Li. Learning n:m fine-grained structured sparse neural networks from scratch. In International Conference on Learning Representations, 2021.

---

> > ### Comment · Reviewer_KcKK · 2021-08-24
> > **Rebuttal Response**
> >
> > Thanks for your detailed response to my review.
> >
> > I was satisfied with your response to my concerns about novelty. No longer needing to pre-train from scratch, insights into how sparsity is induced, and showing that ADMM's hyperparameters require tuning are all valuable contributions. Having read the other reviews and your responses, I am more convinced that the GLUE results are meaningful, given that they were consistent across random seeds and the GLUE benchmark is quite saturated.
> >
> > I have therefore updated my score from 5 to 7.

---

### Official Review · Reviewer_x2gA · 2021-07-18

**Rating:** 7
**Confidence:** 4

**Summary:**

This paper addresses sparsifying Transformers for improving efficiency on NLP tasks. In particular, they use ADMM to alternate between optimizing the weights of the model to perform well on a fine-tuning task, and inducing "NxM" sparsity (out of N contiguous parameters, only M are non-zero).

**Ethical Concerns:**

None.

**Limitations And Societal Impact:**

This paper sufficiently addresses limitations and societal impact.

**Main Review:**

Originality: There is some previous work that aims to induce this kind of semi-structured sparsity using an alternating approach in a neural network. There is much work on inducing a wide variety of sparsity constraints using alternating approaches like ADMM for more classical models (e.g., linear models), and I would recommend the authors include more of these citations in their related work. It is clear how this work differs from previous work, though appropriate baselines are missing.

Quality: The submission is technically sound, this is an appropriate use of ADMM. The datasets evaluated are standard, though have drawbacks; a number of the GLUE tasks are quite small and have high variance even when only varying the random seed, and so the results on validation sets for MRPC, CoLA, etc. should be taken with a grain of salt. The authors claim on line 80 that, "We conduct comprehensive experiments and demonstrate that NxMTransformer achieves 1.7 points higher accuracy than state-of-the-art techniques to induce NxM sparsity", but without more appropriate baselines this claim is not well supported. I don't think this is enough to warrant rejection, but it should be addressed.

Clarity: The paper is well-written. I have a few suggestions for how to make it more clear:
When introducing the decomposition into sub-problems, the doal variable U is introduced, but not described in much detail. Equations 3 and 4 are otherwise fairly clear, though I had to go use external sources to read up on ADMM to remind myself how this worked, and this shouldn't be necessary.
In Table 1, say how sparse the resulting models are.
In Section 4.2, the paragraph lines 252-265, say the dataset(s) being evaluated.
Line 188 is missing the word "task".
Nice job including the information in lines 203-209! That's very helpful.
Line 221 is one of the first times I see it noted that you're aiming for 50% sparsity -- this should be much earlier. If it's already mentioned earlier as well, it should be highlighted more.


Significance: My main complaints are about baselines, and variance from random seeds.
For example, Table 1 doesn't show variance from random seeds, but it should, because the variance for tasks like RTE and MRPC can be larger than the "gains" presented here.
Line 241 claims on the small tasks have the best results, and while it's possible that's a reproducible result, 2.9 points is not a large difference for those tasks.
In Table 1, you bold your "MNLI mm" result, even though it's the same as ASP; this is overclaiming a bit.
The baseline you use in Table 1 is "ASP", and the description on line 217 sounds as though this involves taking a pretrained model, performing one-shot magnitude-based pruning, then fine-tuning the pruned model on a task. To ablate the choice of using the ADMM algorithm, you should also compare against pruning after fine-tuning. Please let me know if I'm understanding this correctly (and make this more clear in the paper). As it stands, you don't have a good baseline justifying why you need to use the more complicated algorithm you introduce.
Figure 2 (b) isn't very informative, you can remove it or put it in the appendix to save space.
Table 2 results are just not informative without a baseline. Your main claim is that your ADMM approach for sparsifying a model is useful even when applied to distilled models, but to support that claim you need to compare against baselines not using ADMM (so, just magnitude pruning before fine-tuning, and magnitude pruning after fine-tuning). As a reader, if I wanted to sparsify a distilled model, Table 2 without a baseline doesn't provide enough evidence for me to choose your ADMM approach over something more simple.

While I do argue for better baselines in this paper, I still think it's a valuable contribution and worth publishing.

**Time Spent Reviewing:**

2

---

> ### Author Response · Authors · 2021-08-10
> **Addressing Small Task Variability**
>
> Thank you for taking the time to review our submission.
>
> As you note, NxMTransformer provides greater accuracy gains across the smaller tasks, which can show higher sensitivity to hyperparameters and random seed. We have performed additional trials across the small configurations and have measured similar sensitivity to random seed for both ASP and NxMTransformer. Across the tasks with fewer than 10,000 training examples, NxM Transformer still provides 1.9 points of accuracy increase on average (1.9% on CoLA, 2.2% MRPC, 3.3% RTE, .3% STS-B) when comparing the respective medians of the random seed sweep. We believe the consistent accuracy gain across this random sweep demonstrates NxMTransformer’s superior accuracy is an artifact of the technique rather than random noise. We will include the results in revision.

---

> > ### Comment · Reviewer_x2gA · 2021-08-13
> > **Small Task Variability**
> >
> > Great, thanks for this. If the gains are as consistent as you describe, then I think there is enough evidence to support the claim that the improvement is from the technique, and not just noise.
> >
> > An example citation for ADMM for sparsity in linear models:
> > Making the Most of Bag of Words: Sentence Regularization with Alternating Direction Method of Multipliers
> > Dani Yogatama, Noah A. Smith
> > ICML 2014

---

### Decision · Program_Chairs · 2021-09-27

**Decision:**

Accept (Poster)

**Comment:**

This paper uses ADMM to induce structured weight sparsity during neural net finetuning.

It is specifically targeting a very local sparsity pattern (N nonzeros in every block of M weights) which is supported in recent nvidia hardware and libraries with near-linear speedups. This is a different constraint scenario than most structured sparsity approaches, and ADMM does seem like a good fit.

As such, reviewers find the contribution valuable but also highlight some weaknesses. The experimental methodology seems susceptible to noise; multiple runs would help; however, simply getting comparable performance seems sufficient when sparsity is the goal. Reviewers and possibly all readers are left wanting speed comparisons; authors should place more emphasis on the library+hardware support for NxM-sparsity to better clarify that this is established independent of this work. The writing and presentation should be polished as well.